# Bis-indole chiral architectures for asymmetric catalysis

Junshan Lai[1], Benjamin List [2]✉ & Jolene P. Reid [1]✉

Chiral scaffolds are essential to the advancement of asymmetric synthesis, yet the development of privileged motifs that more effectively communicate asymmetry constitutes a grand challenge for chemists. Here we describe a method using a confined chiral Brønsted acid catalyst to combine two inexpensive and widely available materials—indole and acetone—into a class of $C_2$-symmetric, spirocyclic compounds called SPINDOLE. SPINDOLEs extend the versatility of established frameworks by offering greater flexibility and ease of synthesis. The resulting chiral compounds can be readily modified to create diverse structures that excel in promoting highly selective reactions such as hydrogenation, allylic alkylation, hydroboration, and Michael addition. This work introduces a powerful strategy for advancing asymmetric catalysis, enabling the creation of versatile chiral frameworks with broad synthetic potential.

1,1'-Binaphthols (BINOLs)[1–3] are widely regarded as "privileged" scaffolds[4,5] in asymmetric catalysis due to their inherent axial chirality, modular design, and straightforward synthesis from inexpensive 2-naphthol[6]. 1,1'-Spirobisindane−7,7'-diols (SPINOLs) represent a conceptual synthetic evolution of BINOLs, incorporating a spirocyclic architecture that offers increased rigidity and stability[7]. These factors render SPINOL as an equally compelling platform for the development of chiral catalysts and ligands, providing solutions to complex problems in asymmetric chemistry[8,9]. Although their performance is notable, the study of SPINOLs has been impeded by their high cost and the paucity of efficient chemical synthesis methods[10]. Fundamental challenges arise in the key synthesis step that involves the two-fold intramolecular Friedel-Crafts alkylation-induced spirocyclization. The inherent site selectivity and low reactivity of this process necessitate additional steps to implement directing group strategies and the use of large amounts of hazardous acids (Fig. 1A). Moreover, the development of sufficiently active chiral catalysts capable of inducing asymmetry at the spiro quaternary carbon stereocenter while tolerating diverse electronic and steric functional groups are rare.

Approaches using asymmetric catalysis are typically lengthy, involving specialized starting materials that necessitate multistep syntheses, complex catalysts, and excessive hazardous reagents (Fig. 1Bi, ii)[11,12]. These factors contribute to significant synthesis costs,

thereby limiting their widespread application. Substrate-controlled enantioselectivity is well precedented in diastereoselective additions to pre-formed chiral ketones (Fig. 1Biii, iv)[13,14]. An alternative approach involves rearranging bisphenols formed via the condensation of phenols and acetone under acidic conditions (Fig. 1C)[15,16]. While the abundant and inexpensive supply of the starting materials render this strategy intuitively attractive, the laborious post-synthesis derivatization and harsh conditions limit its practicality for scalable catalytic integration.

By altering the primary aromatic constituent within the spirocyclic structure, we considered whether these synthetic approaches could be united in a fundamentally unique way. We envisaged that indoles should, in principle, constitute excellent substrates, as the inclusion of the ring nitrogen would be expected to offer simpler syntheses and derivative approaches, while retaining the desirable features for catalysis. To that extent, these heteroaromatic substrates not only react efficiently with acetone but also exhibit inherent site-selectivity that eliminate the need for post-synthesis modifications. During the course of our study, the Tan[17] and Sun[18] group reported strategies as applied to SPINOLs to prepare spiro-bisindoles respectively (Fig. 1D), highly supporting our envisioned approach (Fig. 1E).

In this context, indole has been applied to the Brønsted acid-induced condensation of acetone on a few occasions; however, the

[1]Department of Chemistry, University of British Columbia, Vancouver, BC, Canada. [2]Max-Planck-Institut für Kohlenforschung, Kaiser-Wilhelm-Platz 1, Mülheim an der Ruhr, Germany. ✉e-mail: list@kofo.mpg.de; jreid@chem.ubc.ca

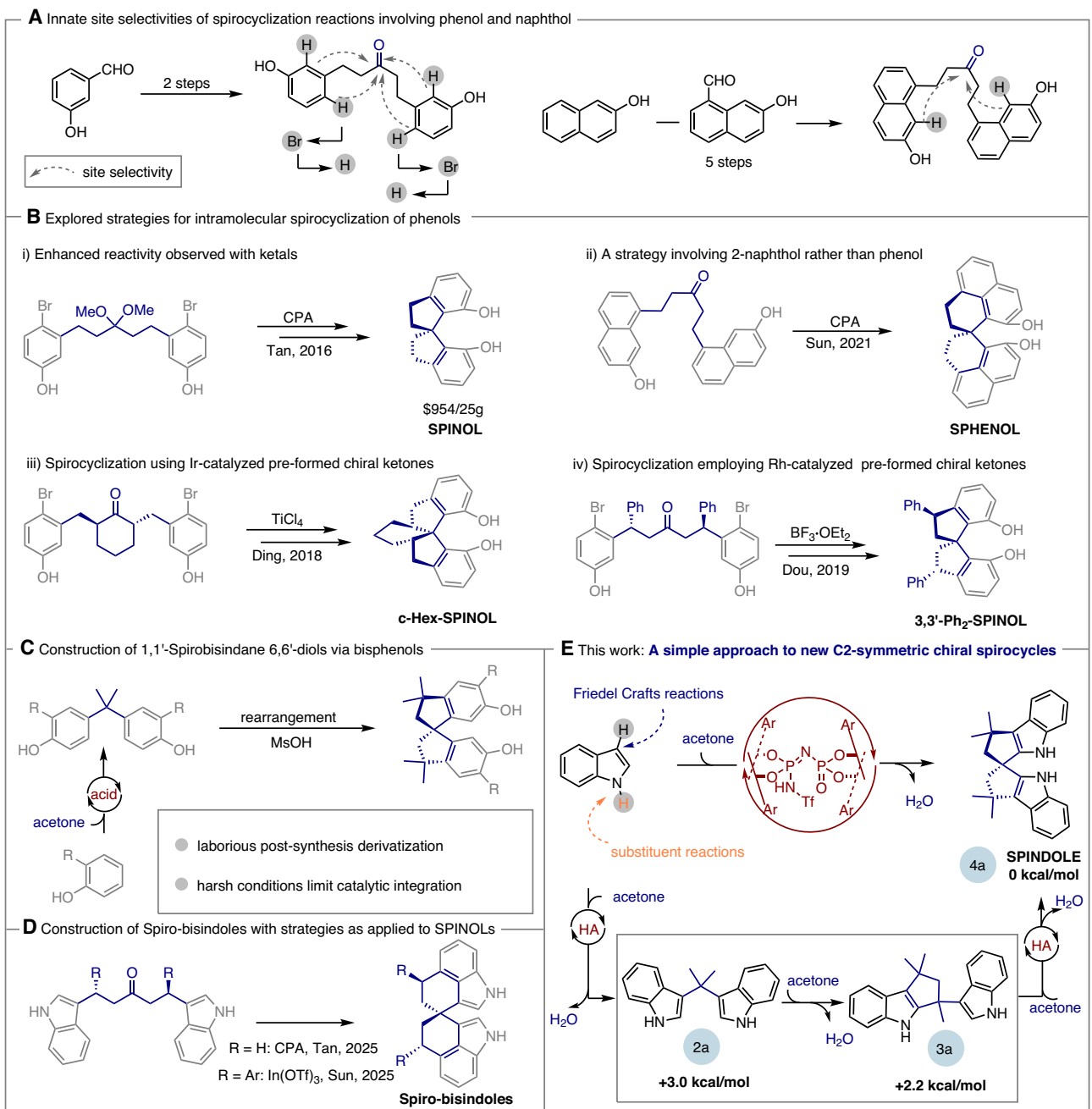

**Fig. 1 | Challenges in the construction of C₂-symmetric axially spirocyclic scaffolds. A** The key synthesis step of forming spirocyclic architectures from phenol and naphthol is fundamentally challenging due to inherent site selectivity and low reactivity. These issues often require additional steps, such as employing directing group strategies and using large amounts of hazardous acids. **B** Strategies to achieve intramolecular spirocyclization of phenols include (i) Enhanced reactivity observed with ketals, (ii) A strategy involving 2-naphthol rather than phenol, (iii) Spirocyclization using Ir-catalyzed pre-formed chiral ketones, and (iv) Spirocyclization employing Rh-catalyzed pre-formed chiral ketones. **C** The construction of 1,1′-spirobisindane-6,6′-diols has also been explored by rearranging bisphenols formed via the condensation of phenols and acetone under acidic conditions. **D** Construction of Spiro-bisindoles with strategies as applied to SPINOLs. **E** In this work, a simpler approach is introduced using a confined chiral Brønsted acid-catalyzed process with readily available and inexpensive indole and acetone. This method provides a direct route to C₂-symmetric spirocycles, addressing the thermodynamic challenges of bis-indole structures, as demonstrated by relative energy data.

---

major products of these reactions are bis(indolyl)propanes[19] and cyclopenta[b]indoles[20,21], with the spirocycle occurring in trace amounts[20]. As little work on this transformation had been previously described, we performed preliminary calculations to recognize any potential challenges that may be responsible for its difficulty. Quantum mechanical (QM) calculations predict that this reaction has a complex thermodynamic profile, with as little as 2.2 kcal/mol separating the bis-indole structures

(see Supplementary pages 4–5 for details). Consequently, selective formation of the desired spirocyclic product requires a catalyst capable of suppressing competing low-energy pathways and destabilizing unwanted intermediates or side products. We reasoned that a chiral catalyst with carefully optimized steric and electronic properties could achieve this by selectively stabilizing the target product and enabling high enantiocontrol through noncovalent interactions (Fig. 1E).

Inspired by the profound structural diversity of the potential product structures ranging from achiral, planar forms to chiral structures with varied spatial configuration and key synthetic studies on chiral Brønsted acids (BAs) in selective catalysis, we initially hypothesized that that phosphoric acids (PAs) with phenyl-derived substituents in the 3,3′-positions, in particular, could offer the necessary shape complementarity to effectively stabilize the targeted structure. Moreover, the extensive availability of established or custom-designed PAs with well-defined chiral pockets provides a robust platform for further catalyst structure optimization[22]. We herein report that these fundamental acid-mediated transformations between indole and acetone can be expanded to produce spirocyclic bis-indoles (hereafter referred to as SPINDOLEs) in high levels of enantioselectivity, enabled by a confined BA catalyst featuring unique aromatic groups.

## Results

### Reaction optimization

Brønsted acids (BAs) are user-friendly catalysts, valued for their stability in the presence of oxygen and water, as well as their long-term storability. From a practical standpoint, BAs are environmentally benign and well-suited for large-scale synthesis, making them an appealing chemotype for exploration[23,24]. Since their pioneering application by Terada and Akiyama[25,26], chiral phosphoric acids (CPAs) have proven to be particularly effective BA catalysts, especially in reactions involving basic electrophiles and protic nucleophiles[27]. CPAs have been widely applied in enantioselective transformations such as Friedel-Crafts alkylations[28,29] and Pictet-Spengler reactions[30], which are particularly relevant to our system under study. Their catalytic effectiveness derives from their ability to form multiple hydrogen bonds with substrates and their modular, tunable frameworks that enable precise enantioselective control in previously challenging reaction spaces. These features, combined with their established success across diverse reaction types, ultimately motivated their use as a starting point for catalyst optimization in this work.

As part of our initial screening efforts, we discovered that the reaction between indole and acetone at 80 °C in toluene with CPA **A7** catalyzed the formation of the targeted spirocycle **4a**, achieving modest yield but encouraging enantioselectivity 56% ee (Fig. 2). To explore the impact of catalyst properties, we evaluated an expanded inventory of CPAs with varied 3,3′-substituents, which influence both steric effects and pKa. A clear trend emerged: stronger CPAs (lower pKa values) improved transformation efficiency (see Supplementary Fig. S1). However, the formation of the by-product cyclopenta[b]indole **3a** occurred to varying degrees and appeared unavoidable with this catalyst class. Notably, CPA **A14**, previously shown by Tan and Sun to deliver optimal performance for the asymmetric synthesis of SPINOL[11] and SPHENOL[12], did not exhibit efficient reactivity or selectivity in our system. Marked improvements in yield were obtained using N-triflyl phosphoric acids (NPAs), which possess lower pKa values[31]. Under these stronger acidic conditions, the undesired cyclopenta[b]indole **3a** was virtually eliminated, although enantioselectivity did not improve. This outcome suggests that the higher proton availability of NPAs plays a critical role in driving spirocycle **4a** formation. Ultimately, selective access to either cyclopenta[b]indole **3a** or targeted spirocycle **4a** can be effectively controlled by tuning the Brønsted acid structure based on pKa.

Given the limitations in enantiofacial discrimination with CPAs and NPAs, we turned to more confined Brønsted acids—specifically imidodiphosphoric acids (IDP)[32], iminoimidodiphosphoric acids (iIDP)[33], and imidodiphosphorimidates (IDPi)[34,35]. These catalysts, characterized by their low pKa values and sterically encumbered active sites, have been shown to enhance both reactivity and enantioselectivity in asymmetric transformations, offering catalytic sites within chiral microenvironments that enhance both reactivity and enantioselectivity in asymmetric transformations[36]. Through extensive evaluation of these catalyst classes, we identified the iIDP catalyst **D4** as optimal, achieving 90% ee and 94% yield at 60 °C in toluene. Further optimization led to the following conditions: treating **1a** (0.2 mmol) with acetone (5.0 equiv.) in the presence of **D4** (2.5 mol%) in THF (0.2 M) at 60 °C for 5 days yielded **4a** with 86% yield and 96% ee. Excess acetone was essential to drive the reaction to completion, as it serves as the electrophile in the sequential Friedel−Crafts alkylation leading to the spirocyclic framework. The $C_{10}F_7$ group in the catalyst is crucial for the enantiocontrol, as the $C_{10}F_7$-substituted catalyst (**D4, E4**) significantly boosted the enantioselectivity from 0−7% ee to 80−90% ee when comparing to the analogous catalysts (**D1-D3, E1-E3**). Notably, **D4** was synthesized via a palladium-free route, incorporating a 3,3′-

**Fig. 2 | Optimization of chiral Brønsted acid catalyst structures.** The reaction of **1a** (0.1 mmol), acetone (0.5 mmol), and stated amount of catalysts was carried out in 0.5 ml of toluene in a 4 mL heavy wall vessel. The vessel was sealed and stirred at the stated temperature for 3 days (more details see Supplementary Figs. S1−S5). Isolated yields are shown. The ee values were determined by chiral SFC analysis.

| | CPA | IDP | iIDP | NPA | IDPi |
|---|---|---|---|---|---|
| pKa (in MeCN) | $pK_a \sim 13$ | $pK_a \sim 11$ | $pK_a \sim 9$ | $pK_a \sim 6$ | $pK_a \sim 4$ |
| Yield (**4a**) | 17-66% yield | 50-90% yield | 75-95% yield | 72-90% yield | 85-92% yield |
| ee up to (**4a**) | 56% ee | 80% ee | 90% ee | 15% ee | 14% ee |

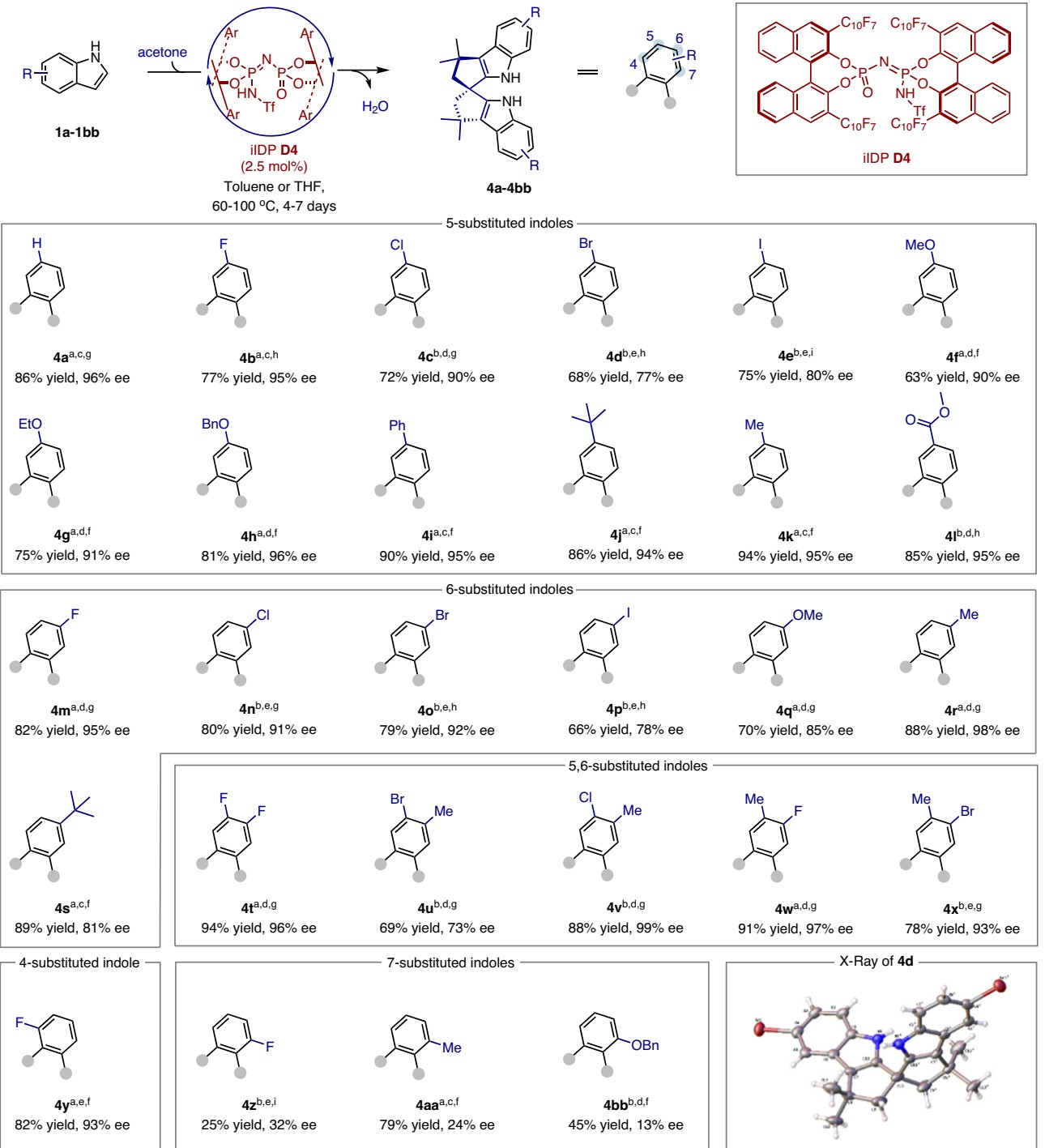

**Fig. 3 | Various indole substrates tested with the optimal conditions.** Condition: **1a-1bb** (0.2 mmol, 1.0 eq), acetone (1.0 mmol, 5.0 eq) iIDP **D4** (9.6 mg, 0.005 mmol, 2.5 mol%), toluene or THF (1.0 mL), stirred at indicated temperature for the indicated days. Isolated yields given. Enantioselectivities (ee) were measured by SFC. Absolute configurations confirmed by the X-ray crystallographic analysis after recrystallization of **4d**. The stereochemistry of the remainder of the entries is assigned by analogy. (a) Reactions were conducted in toluene. (b) Reactions were conducted in THF. (c) Reactions were conducted at 60 °C. (d) Reactions were conducted at 80 °C. (e) Reactions were conducted at 100 °C. (f) Reactions were conducted for 4 days. (g) Reactions were conducted for 5 days. (h) Reactions were conducted for 6 days. (i) Reactions were conducted for 7 days.

$C_{10}F_7$ group on the BINOL backbone[37]. This design offers a sustainable and cost-effective catalytic process, with water as the sole by-product.

## Substrate scope
With the optimized conditions in place, we evaluated a broad range of indoles with varying electronic properties and ring substitutions

(Fig. 3). The transformation showed compatibility with C5 and C6 substitutions on the indole ring, yielding the desired spirocyclic products (**4a–4y**) in moderate to good yields (63–94%) with high enantioselectivity (up to 99% ee). Electron-rich, electron-deficient, and sterically demanding substituents—including alkyl, alkoxy, halides, and ester groups—at C5 or C6 were well-tolerated, consistently delivering

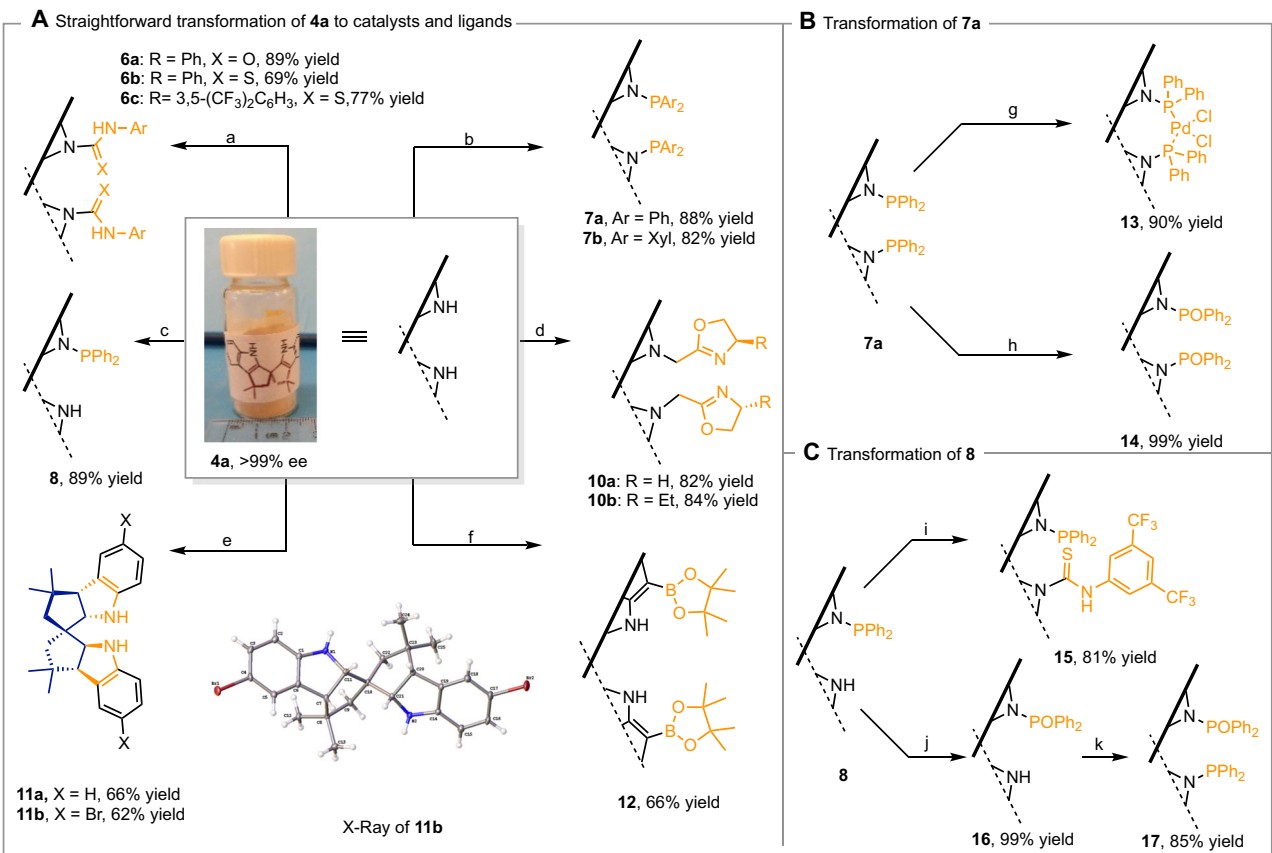

**Fig. 4 | Elaboration of SPINDOLE to generate a diverse set of chiral ligand and organocatalyst structures. A** Straightforward transformation of **4a** to catalysts and ligands. Conditions: (a) **4a** (1.0 eq), NaH (3.0 eq), DMF, 0 °C, 1 h; PhNCO (**5a**)/ PhNCS (**5b**)/3,5-(CF₃)₂C₆H₃-NCS (**5c**) (3.0 eq), rt, 16 h. (b) **4a** (1.0 eq), *n*BuLi (2.2 eq), THF, 0 °C, 1 h; Ph₂PCl or Xyl₂PCl (2.2 eq), rt, 16 h. (c) **4a** (1.0 eq), *n*BuLi (1.1 eq), THF, −78 °C, 1 h; Ph₂PCl (1.1 eq), rt, 2 h. (d) **4a** (1.0 eq), NaH (3.0 eq), DMF, 0 °C, 1 h; 2-(chloromethyl)-4,5-dihydrooxazole (**9a**) or (*R*)-2-(chloromethyl)-4-ethyl-4,5-dihydrooxazole (**9b**) (3.0 eq), rt, 16 h. (e) **4a** or **4 d** (77% ee) (1.0eq), NaCNBH₃ (10. 0 eq),

HOAc (10.0 eq), DCM, 0 °C, 4 h. (f) **4a** (1.0eq), B₂pin₂ (2.0 eq), [Ir(cod)(OMe)]₂ (1.5 mol%), dtbpy (3.0 mol%), THF, reflux, 24 h. **B** Transformation of **7a**. Conditions: (g) **7a** (1.0 eq), Pd(PhCN)₂Cl₂ (1.0 eq), Benzene, 16 h. (h) **7a** (1.0 eq), H₂O₂ (10.0 eq), DCM, rt, 1 h. **C** Transformation of **8**. Conditions: (i) **8** (1.0 eq), NaH (2.0 eq), DMF/ THF (v/v = 1/4), 0 °C, 0.5 h; 3,5-(CF₃)₂C₆H₃-NCS (**5c**) (2.0 eq), rt, 3 h. (j) **8** (1.0 eq), H₂O₂ (5.0 eq), DCM, rt, 1 h. (k) **16** (1.0 eq), *n*BuLi (1.1 eq), THF, −78 °C, 1 h; Ph₂PCl (1.1 eq), rt, 16 h.

products with 90–99% ee (e.g. **4a–4c**, **4f-4o**, **4q–4x**). In addition, we observed that indoles bearing electron-deficient or bulky substituents faced challenges in reactivity and selectivity. Specifically, substrates with -Br, -I, or bulky groups (**4d–4e**, **4p–4q**, **4s**, **4u**) achieved enantioselectivities of 73–85% ee. Meanwhile, electron-deficient indoles (-Cl, -Br, -I, -COOMe) exhibited reduced reactivity under our conditions, necessitating the use of toluene as a solvent and elevated temperatures for effective transformation. Substrates with -CF₃ or -CN groups failed to yield the desired products, while acyl and -CH₂OH substituents led to polymerization (Supplementary Fig. S10). We anticipated that substituents near the reaction site, particularly at the functional group binding to the catalyst, might alter the preferred substrate conformation, potentially affecting key interactions with the chiral catalyst. Halides (-Cl, -Br, -I) at C7 (Supplementary Fig. S10) did not yield desired products, even at elevated temperatures, while -Me and -COOMe at C4 (Supplementary Fig. S10) showed no conversion, with the starting material recovered. Electron-donating groups (-Me, -OBn) at C7 (**4z–4bb**) yielded low to moderate conversion with enantioselectivities of 24% and 13% ee, respectively. Encouragingly, indole bearing a -F group at C4 (**4y**) provided the desired product with a high enantioselectivity (82% yield, 93% ee). Interestingly, multisubstituted indoles at C5 and C6 consistently delivered exceptional results, achieving up to 99% ee (**4t-4x**), representing some of the highest enantioselectivities reported for confined catalysts in asymmetric transformations (99% ee, **4v**).

To demonstrate the practicality and scalability of this protocol, preparative synthesis of **4a** was carried out on a 40 mmol (4.68 g) scale of indole. The desired product, **4a**, was obtained in 84% yield (5.94 g) with 90% ee after 5 days of stirring in toluene. Enantiopure (*R*)-**4a** could be easily obtained by recrystallization in Et₂O and hexanes, with 88% recovery and >99% ee. The opposite enantiomer, (*S*)-**4a**, was achieved with similar results by using (*S,S*)-**D4** as the catalyst.

**Chiral ligand and catalyst development**

As noted above, the conformational rigidity, chemical stability, and modularity of axially spirocyclic scaffolds render them important backbones for chiral catalysts and ligands. Building on these advantages, the developed indole-modified spirocyclic scaffold **4a** not only offers a straightforward synthesis approach using inexpensive indole and acetone as starting materials but also introduces milder and simpler derivatization methods for constructing diverse and potentially useful catalyst and ligand frameworks, comparable to BINOL and SPINOL analogs (Fig. 4). The weakly acidic NH functionality facilitates various substituent reactions through cost-effective, noble-metal-free method, which is particularly important in the cases of phosphine ligands[38]. This modularity is demonstrated by the synthesis of diverse compounds, including chiral urea (**6a**), thiourea (**6b**, **6c**) bearing aromatic groups, and dihydrooxazoles (**10a**, **10b**), which were obtained in high yields through efficient single-step reactions. Bisphosphines (**7a**, **7b**) and monophosphine (**8**) were obtained by controlling the amount

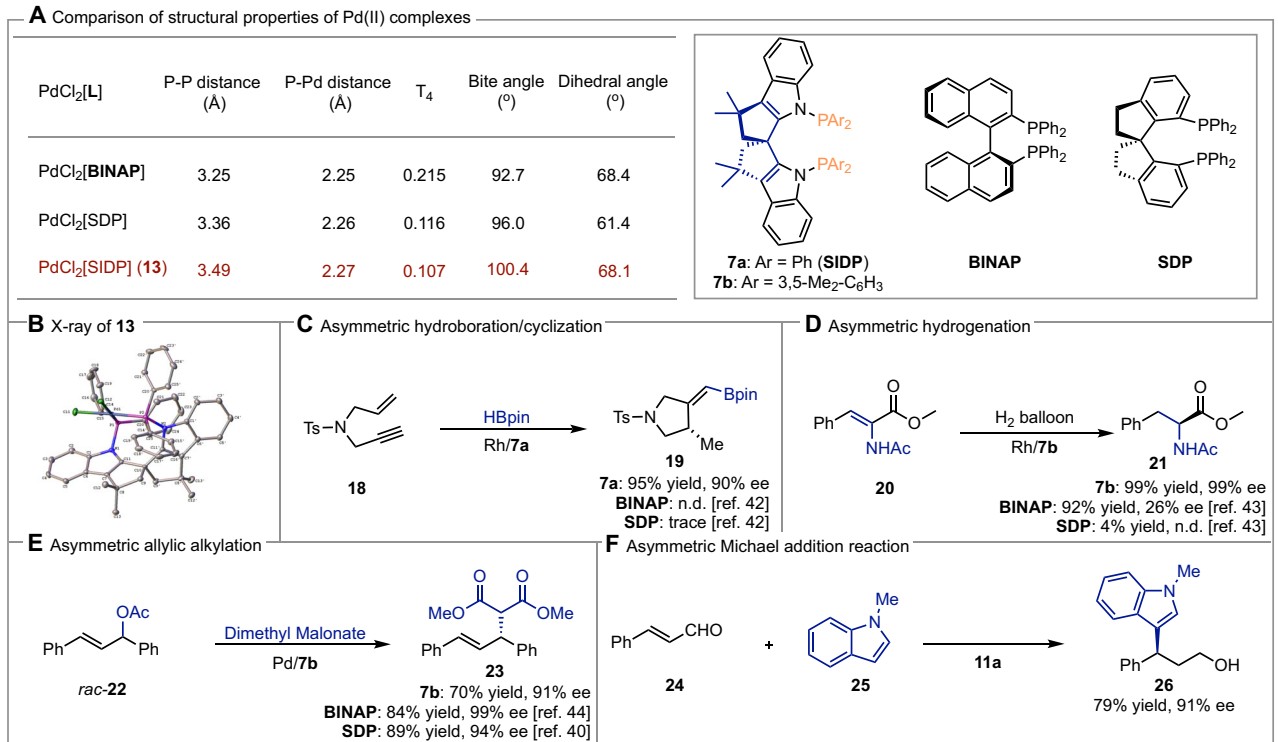

**Fig. 5 | Structural properties of the Pd(II) complex and synthetic applications of the derived ligands and organocatalyst. A** Comparing structural properties of Pd(II) complexes. **B** X-ray of Pd(II) complex **13**. **C** Rhodium/**7a** catalyzed asymmetric hydroboration/cyclization. Condition: **18** (1.0 eq), HBpin (1.5 eq), Rh(cod)$_2$BF$_4$ (1.0 mol%), SIDP **7a** (1.2 mol%), DCM, 0 °C, 24 h. (**D**) Rhodium/**7b** catalyzed asymmetric hydrogenation. Condition: **20** (1.0 eq), Rh(cod)$_2$SbF$_6$ (1.0 mol%), SIDP **7b** (1.2 mol%), H$_2$ balloon, DCM, rt, 6 h. **E** Palladium/**7b** catalyzed asymmetric allylic alkylation. Condition: *rac*-**22** (1.0 eq), Dimethyl Malonate (2.0 eq), [Pd(allyl)Cl]$_2$ (2.5 mol%), SIDP **7b** (6 mol%), ZnEt$_2$ (2.0 eq), THF, 0 °C, 24 h. (**F**) Secondary amine **11a** catalyzed asymmetric Michael addition reaction. Condition: (1) **24** (1.0 eq), **25** (2.0 eq), **11a** (5 mol%), TFA (10 mol%), DCM/IPA, 0 °C, 48 h; (2) NaBH$_4$, MeOH, rt, 10 min.

of nBuLi and the reaction temperature, and these were easily converted to corresponding phosphites (**14**, **16**) by H$_2$O$_2$ oxidation. The monophosphine scaffold can be further diversified with various N1,1'-substituents (**15**, **17**), underscoring the scaffold's flexibility. Additionally, **11a** was synthesized by reducing **4a** with NaCNBH$_3$, and can be evaluated as a secondary amine catalyst. The stereochemistry of **11a** was assigned based on its analog **11b**, confirmed by X-ray crystallography. Furthermore, the NH groups of the indole rings function as directing groups in Ir-catalyzed C7,7'-diborylation (**12**), demonstrating potential for diversifying the C7,7'-positions of SPINDOLEs and addressing limitations of C7-substituted indoles. Notably, none of these processes showed any evidence of racemization, highlighting the robustness of the approach.

Building on the success of using a single structural scaffold to generate a diverse range of functionalized components, we were compelled to investigate whether these tailored structures could extend effectively into catalytic applications. The versatility observed in structural modification underscores the potential of this scaffold as a platform for both ligand development and organocatalysis. With this in mind, we aimed to assess whether the achieved structural diversity could translate into meaningful catalytic activity, establishing these compounds as versatile tools for asymmetric synthesis.

Our initial focus was on bisphosphine derivatives due to their established importance in catalysis, particularly in stabilizing palladium (II) complexes. Given the distinct structural features of our scaffold (e.g. P-P distance, τ4, Bite angle, more details see Supplementary Table S3), we anticipated that it might exhibit unique properties relative to widely used BINAP[39] and SDP[40] ligands (Fig. 5A). As a preliminary assessment, we conducted a detailed comparison of key structural parameters—such as dihedral angles, $^{31}$P NMR shifts, and bite

angles—to understand how our scaffold fits within established chemical space. If our structures demonstrate unique properties through these metrics, they could hold significant promise for producing reaction outcomes in organic synthesis.

To this end, we obtained the crystal structure of the palladium complex **13** and compared it to reported BINAP- and SDP-coordinated palladium complexes (Fig. 5B). Notably, the bite angle (P-Pd-P) in PdCl$_2$(SIDP) **13** complex is 100.4°, substantially larger than that in PdCl$_2$(BINAP) (92.7°) and PdCl$_2$(SDP) (96.0°), with a correspondingly longer P-P distance (3.49 Å) while maintaining similar Pd-P bond lengths. This increased bite angle likely reflects the inherent rigidity of our scaffold, potentially imparting unique steric and electronic properties in catalytic applications.

Further analysis showed that the dihedral angle between the phosphine phenyl rings in PdCl$_2$(SIDP) (68.1°) closely resembles that in PdCl$_2$(BINAP) (68.4°) but is significantly larger than in PdCl$_2$(SDP) (61.4°). Additionally, PdCl$_2$(SIDP) **13** has a lower geometry index parameter ($\tau_4 = 0.107$) compared to BINAP and SDP analogs, indicating a less skewed square-planar geometry, which may influence reactivity and selectivity in asymmetric reactions. DFT analysis and $^{31}$P NMR studies further underscore these structural differences, likely due to the N1,1' configuration of the indole backbone (see Supplementary Table S3 for details). Moreover, the NH bonds in SPINDOLE are positioned closer to the spirocenter compared to SPINOL (2.65 Å vs. 3.04 Å, as measured from X-ray crystallographic structures of **4d** and SPINOL[41]), potentially resulting in a deeper active site for catalysis.

The distinct structural features of our scaffold raise intriguing questions about its performance in catalytic applications, especially in comparison to traditional ligands like BINAP and SDP. Motivated by these differences, we first evaluated compound **7a** as a ligand in the

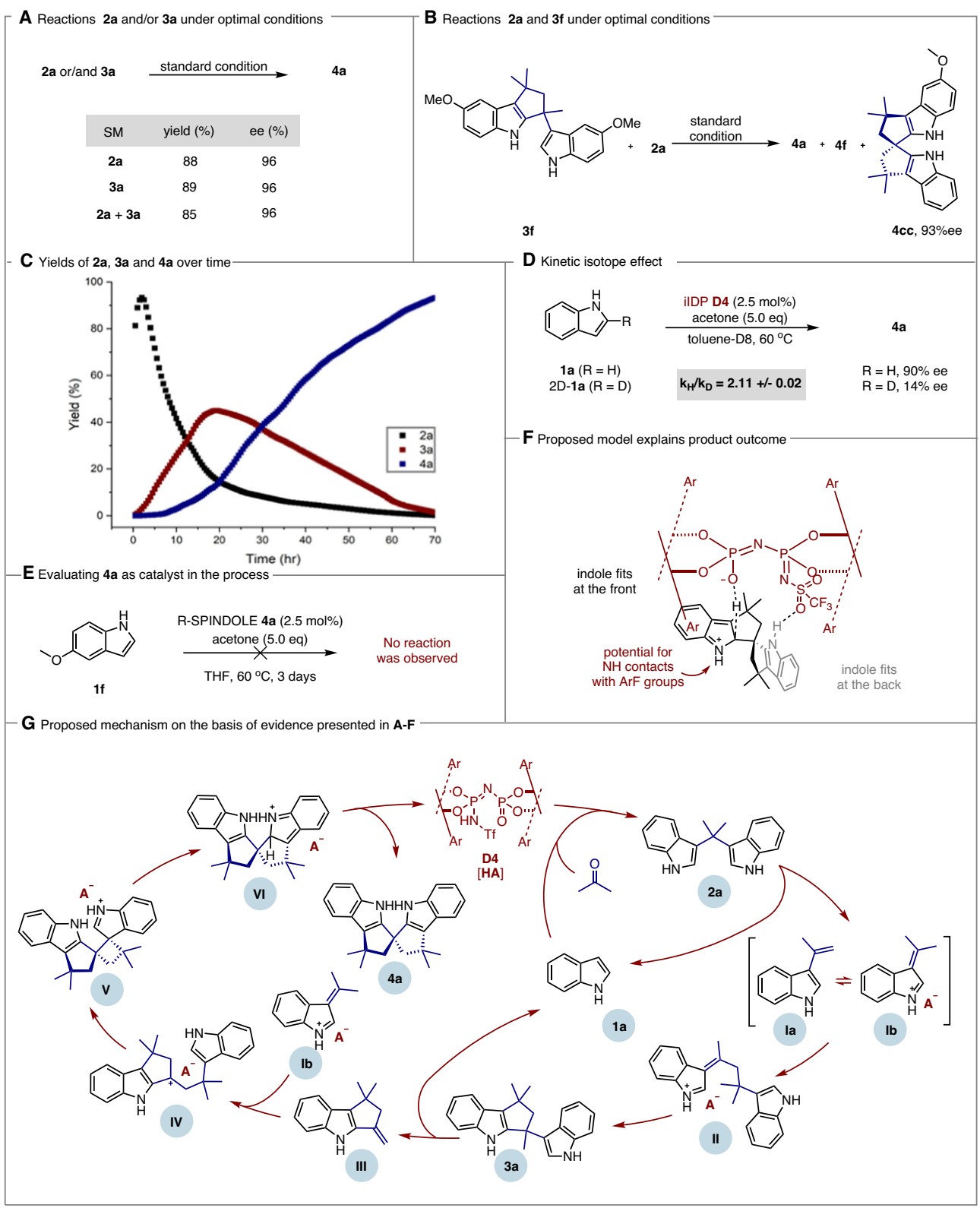

**Fig. 6 | Mechanism studies. A** Results of performing reactions with intermediates **2a** and/or **3a** under optimal conditions. **B** Positive cross over demonstrates reversibility. **C** Monitoring the concentration of **2a**, **3a** and **4a** over time. **D** Experiment reads out primary kinetic isotope effect. **E** Evaluating SPINDOLE **4a** catalytic reactivity. **F** Proposed stereochemical model for explaining the selectivity determined at the deprotonation step. **G** Proposed mechanism on the basis of evidence presented in A-F.

Rh-catalyzed asymmetric hydroboration/cyclization of 1,6-enyne. Remarkably, the desired product was obtained in 90% ee with a 95% yield (Fig. 5C), while both BINAP and SDP showed limited reactivity and selectivity in similar reactions[42]. In the Rh-catalyzed asymmetric

hydrogenation of **20** (Fig. 5D), we were delighted to observe 99% ee for product **21** when using **7b** as the ligand. In contrast, the same reaction with BINAP resulted in low enantioselectivity and barely proceeded with SDP, as referenced in reported studies[43]. We further assessed our

scaffold in a Pd-catalyzed asymmetric allylic alkylation reaction (Fig. 5E), where **7b** again performed well, achieving 91% ee—comparable to SDP[44] and BINAP[40].

At this stage, we envisaged that a compelling demonstration of the potential of these structures would be to successfully apply it to a different class of compound entirely. Given the broad utility of secondary amine compounds in catalysis[45], we considered the asymmetric Michael addition of cinnamaldehyde **24** and N-methylindole **25** to be of significant practical value (Fig. 5F). Using TFA as a co-catalyst of **11a**, we achieved 91% ee, underscoring the scaffold's versatility in catalysis. These promising results highlight the strong potential of SPINDOLE-based catalysts and ligands in asymmetric transformations. The unique structural features of this scaffold can indeed translate into significant catalytic advantages across diverse reaction types.

### Mechanism studies

The optimization process yielded significant mechanistic insights through the isolation of intermediates **2a** and **3a** at different conversion levels. Subjecting **2a** and racemic **3a** to standard conditions in separate experiments produced the same ee and comparable yields (Fig. 6A). Combining **3f** and **2a** under standard conditions yielded a mixture of **4a**, **4f**, and **4cc** (Fig. 6B), suggesting that the steps preceding cyclopenta[b]indole formation, specifically those involving **3a** and **3f**, are reversible. Although these intermediates we isolate are not necessarily on the most productive pathway, the reversibility of many steps in the process is evident. Stacked $^1$H NMR spectra of the reaction in toluene-d8 revealed that compound **2a** was rapidly formed and consumed in the initial stages, followed by the generation of **3a** and the gradual emergence of the desired spirocycle **4a**. These observations suggest that intermediates **2a**, **3a**, and **4a** are formed and possibly converted sequentially (Fig. 6C).

To elucidate the enantioselectivity-determining step, several careful experiments were conducted. A primary kinetic isotope effect (KIE) of 2.11 was measured using **1a** and its deuterated analogue, 2D-**1a**, as probes. This suggests a potentially complex mechanism in which the spirocyclization step may be reversible. The relative energies of diastereomeric intermediates and the barriers to their deprotonation likely both contribute to the excellent observed enantioselectivities (Fig. 6D). Notably, the use of 2D-**1a** led to a significant decrease in enantioselectivity to 14% ee, supporting the notion that deprotonation is the most likely enantioselectivity-determining step.

Given the structural similarity between SPINDOLE and the catalyst, we explored whether the product could influence the stereochemistry-determining step. To test this, we replaced iIDP with SPINDOLE under otherwise identical conditions. The absence of catalytic activity confirmed that SPINDOLE does not participate in the reaction (Fig. 6E). Additionally, a linear relationship between the enantiopurity of the catalyst and the product was observed, suggesting that a single iIDP molecule is involved in the enantioselectivity-determining step (see Supplementary Fig. S9).

To rationalize the observed enantioselectivity, density functional theory (DFT) calculations were performed at the IEFPCM(THF)-ωB97XD/6-31G(d,p)//ωB97XD/6-31G(d) level of theory on diastereomeric SPINDOLE-iIDP complexes involving **4a** and **D4**. The lowest-energy ground-state complex aligns with the experimentally determined enantiomer, suggesting that the stabilizing interactions in this structure are also likely relevant in the preceding transition state, consistent with the Hammond postulate. Notably, the calculations highlight key interactions between the aromatic groups on the catalyst and the substrate as likely determinants of enantioselectivity during the deprotonation step (see Supplementary pages 5–8 for details). Ultimately, these calculations provided the basis for constructing a preliminary mechanistic model that connects catalyst-intermediate interactions with the stereochemical outcome as shown in Fig. 6F.

Based on experimental results, we propose the following mechanism for the chiral Brønsted acid–catalyzed formation of the axially chiral spirocycle, SPINDOLE (Fig. 6G). The reaction begins with the protonation of acetone by catalyst **D4**, inducing the condensation of indole (**1a**) to form bis(indolyl)propane (**2a**). Under acidic conditions, the indole nitrogen facilitates C(sp³)-C(indolyl) bond cleavage, generating alkene-substituted indole species **Ia** and **Ib**. These intermediates react to form carbocation **II**, which cyclizes to form indolyl-cyclopenta[b]indole (**3a**). Upon heating, **3a** rearranges into alkene species **III**, which reacts with **Ib** to form carbocation **IV**. This intermediate undergoes cyclization, followed by a Pictet-Spengler-type migration and deprotonation, producing the desired product (**4a**) and regenerating **D4**.

### Discussion

In summary, we have developed SPINDOLE, a versatile chiral framework that creates opportunities in asymmetric synthesis. This approach leverages commercially available, inexpensive indoles and acetone to provide an efficient and cost-effective route to enantiopure indole-based axially spirocyclic scaffolds. The methodology offers rapid access to SPINDOLEs in good to excellent yields (up to 94%) and exceptional enantioselectivities (up to 99% ee), demonstrating broad functional group compatibility. Encouraged by these promising results, we have initiated the construction of a comprehensive catalyst and ligand library based on this innovative axially spirocyclic backbone. The SPINDOLE scaffold presents significant potential as a robust platform for the design of organocatalysts and ligands, serving as a complementary alternative to the widely used BINOL and SPINOL frameworks. Ongoing studies in our laboratory are focused on further exploring the utility of SPINDOLE as an axially chiral ligand and organocatalyst, with findings to be reported in future publications.

### Methods

#### General procedure for enantioselective reactions

A 4 mL was charged with 1.0 mL of toluene or THF, **1a-1bb** (0.2 mmol, 1.0 eq), acetone (58.0 mg, 1.0 mmol, 5.0 eq), and iIDP **D4** (9.1 mg, 0.005 mmol, 2.5 mol%) were added. The vessel was sealed and stirred at 60 °C to 100 °C for 4–7 days. The reaction mixture was then subjected to silica gel column chromatography (Petroleum ether: AcOEt = 20:1 to 5:1) to afford compound **4a-4bb**.

### Data availability

All data are available in the main text or the supplementary information (including computational details, experimental details, NMR data, SFC and NMR spectrums). Source data are provided with this paper. All data are available from the corresponding author upon request. The X-ray crystallographic coordinates for structures reported in this study have been deposited at the Cambridge Crystallographic Data Centre (CCDC), under deposition numbers CCDC-2408120 for **4d**; CCDC-2408122 for **11b**; CCDC-2408121 for **13**. These data can be obtained free of charge from The Cambridge Crystallographic Data Centre via www.ccdc.cam.ac.uk/data_request/cif. Source data are provided with this paper.

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

## Acknowledgements

Financial support to J.P.R. was provided by the University of British Columbia, the Natural Sciences and Engineering Research Council of Canada (NSERC) and the CFI John R. Evans Leaders Fund. Computational resources were provided from the Digital Research Alliance of Canada and the Advanced Research Computing (ARC) center at the University of British Columbia. We thank Brian Patrick (UBC) for solving the crystal structure. We thank Professor Jonathan Goodman (University of Cambridge) for insightful discussions regarding the proposed mechanism.

## Author contributions

J.P.R and J.L. performed the calculations. J.L. performed the experiments. B.L. supported the catalyst optimization process. J.P.R and J.L. wrote the manuscript. J.P.R. supervised the project and acquired project funding.

## Funding

## Competing interests

The authors declare no competing interests.
