## [Transparent Peer Review file · Nature Communications]

SPIROCYCLIC BIS-INDOLE CHIRAL ARCHITECTURES FOR ASYMMETRIC CATALYSIS

Corresponding Author: Dr Jolene Reid

Version 0:

Reviewer comments:

Reviewer #1

(Remarks to the Author)

In this manuscript, P. Reid and List proposed a novel chiral spirocyclic bisindole (SPINDOLE) structure and successfully synthesized it using simple starting materials. These findings suggest that the SPINDOLE framework has broad potential applications in asymmetric catalysis. A plausible reaction mechanism was proposed based on a series of control experiments and DFT calculations. The manuscript appears suitable for publication in Nature Communications after addressing the following concerns:

- In Fig. 5, the study highlights the excellent performance of SPINDOLE in various reactions. To better demonstrate its advantages, it is recommended to include previous results with traditional chiral catalysts such as BINOL and SDP ligands.
- While preliminary mechanistic investigations and DFT calculations have been performed, further analysis of catalyst-substrate interactions in the transition states using DFT calculations would provide deeper mechanistic insights into the enantioselectivity.

Reviewer #2

(Remarks to the Author)

This manuscript reports a novel class of C₂-symmetric axially chiral spirocyclic structure termed 'SPINDOLEs', the reaction development on their catalytic asymmetric synthesis, and their applications in asymmetric catalysis. The spirocyclic structure developed here is new, and the structural properties of corresponding ligands distinct from the BINOL or SPINOL-based ones, underscoring their great potentials in catalytic applications. The synthetic method is practically useful, as readily available starting materials and an easily accessible catalyst are employed and the reaction is scalable. The work is further strengthened by detailed mechanistic studies and several applications of the developed structure. Overall, this is an impressive work with high novelty as well as great potential utilities, so it is recommended for publication in Nat. Commun.

A minor point: The C10F7 group in the catalyst is crucial for the enantiocontrol, as the C10F7-substituted catalyst (D4, E4) significantly boosted the enantioselectivity from 0-7% ee to 80-90% ee when comparing to the analogous catalysts (D1-D3, E1-E3). Comments on these key results might be added in the manuscript.

Reviewer #3

(Remarks to the Author)

Chiral scaffolds are indispensable for the progress of asymmetric synthesis, yet the creation of novel, privileged motifs that can more effectively convey asymmetry remains a formidable challenge for chemists. In this manuscript, the authors introduce SPINDOLE, a groundbreaking and versatile chiral framework that ushers in new possibilities for asymmetric synthesis. This innovative approach utilizes readily available and inexpensive indoles and acetone to offer an efficient and cost-effective pathway to enantiopure, axially spirocyclic scaffolds based on indoles. The methodology enables rapid access to SPINDOLEs with good to excellent yields and exceptional enantioselectivities, while demonstrating remarkable compatibility with a wide range of functional groups. Encouraged by these promising results, the authors have embarked on the development of a comprehensive library of catalysts and ligands based on this unique axially spirocyclic backbone. The SPINDOLE scaffold holds significant potential as a robust platform for designing organocatalysts and ligands, offering a

complementary alternative to the widely used BINOL and SPINOL frameworks. Overall, this work presents a powerful strategy for advancing asymmetric catalysis by enabling the creation of versatile chiral frameworks with broad synthetic potential. I recommend the publication of this manuscript in Nature Communications after addressing the following minor issues.

1. As shown in Fig. 3, indole with a -F group at C4 delivers the desired product 4y with high enantioselectivity. In contrast, indole with a -F group at C7 produces the desired product 4z with low enantioselectivity and yield. What could account for this difference?

2. Please re-draw the structures of products 4u-4x (the Cl, Br, and Me groups are not in the proper positions).

3. The Tan's work related to asymmetric synthesis of spiro-bisindoles should be highlighted in the introduction with a dedicated figure, rather than merely being cited in reference 13.

4. The footnote of Figure 4 contains numerous formatting errors. Please address and correct these issues.

Reviewer #4

(Remarks to the Author)

In the present manuscript, Lai, List, and Reid report an efficient method for the catalytic asymmetric synthesis of spirocyclic bis-indole compounds using indoles and acetone as simple starting substrates. The reaction system, conditions, and chiral Brønsted acid organocatalysts were carefully optimized to afford the target spiro bis-indole products with high enantioselectivities and good yields. The multistep reactions were well-controlled using a BINOL-derived iminoimidodiphosphoric acid catalyst.

Furthermore, the derivatization and application of spiro bis-indole derivatives as chiral ligands were demonstrated. The excellent performance of spiro bis-indole-derived chiral phosphine ligands was elegantly showcased in transition-metal-catalyzed asymmetric transformations. Additionally, a spiro bis-indole-derived chiral amine compound proved to be an effective organocatalyst in asymmetric conjugate addition.

Mechanistic studies were also conducted to elucidate the reaction mechanism of this catalytic asymmetric method for the synthesis of spiro bis-indoles. The manuscript is well-organized, and both the main text and supporting information are meticulously prepared. Key references are appropriately introduced and cited. For these reasons, the reviewer recommends that this manuscript be published in Nature Communications without modifications.

Minor suggestion: A very recent publication on the asymmetric synthesis of related spiro bis-indole compounds should be cited: *Angew. Chem. Int. Ed.* 2025, e202424773.

Version 1:

Reviewer comments:

Reviewer #1

(Remarks to the Author)

I agree with the authors that the detailed mechanistic study is rather complicated, and should be addressed in a dedicated mechanistic study. I thus support the publication of the manuscript in its current form.

Reviewer #3

(Remarks to the Author)

The authors have addressed most of points raised. I would like to recommend the publication of this manuscript in Nature Communications without further revision.

Reviewer #1 (Remarks to the Author):

In this manuscript, P. Reid and List proposed a novel chiral spirocyclic bisindole (SPINDOLE) structure and successfully synthesized it using simple starting materials. These findings suggest that the SPINDOLE framework has broad potential applications in asymmetric catalysis. A plausible reaction mechanism was proposed based on a series of control experiments and DFT calculations. The manuscript appears suitable for publication in Nature Communications after addressing the following concerns:

- In Fig. 5, the study highlights the excellent performance of SPINDOLE in various reactions. To better demonstrate its advantages, it is recommended to include previous results with traditional chiral catalysts such as BINOL and SDP ligands.

Response: Thanks for the suggestions, we have updated Fig. 5 as recommended, please see new Fig. 5 (also shown below).

- While preliminary mechanistic investigations and DFT calculations have been performed, further analysis of catalyst-substrate interactions in the transition states using DFT calculations would provide deeper mechanistic insights into the enantioselectivity.

Response: We completely agree that transition state modeling would provide valuable mechanistic insights. However, this level of analysis is outside the construct we chose for this paper, which is primarily synthetic with supporting mechanistic investigations rather than an exhaustive computational study.

Studying transition states in this system presents significant challenges due to the lack of C₂ symmetry, leading to multiple unique binding modes that complicate computational modeling. Additionally, given our preliminary findings on the role of attractive NCIs, high-level methods with dispersion corrections

are required for accurate modeling. As a result, methods like ONIOM calculations are not suitable for capturing these interactions effectively.

We are actively planning a comprehensive study to investigate these effects further, particularly regarding catalyst sensitivity and the unexplained mechanistic observations, such as the role of water in ee erosion and the unexpected enantiomeric enrichment under anhydrous conditions. However, given the complexity of these investigations, we believe they are best addressed in a dedicated mechanistic study rather than within the scope of this paper.

Reviewer #2 (Remarks to the Author):

This manuscript reports a novel class of C₂-symmetric axially chiral spirocyclic structure termed ‘SPINDOLEs’, the reaction development on their catalytic asymmetric synthesis, and their applications in asymmetric catalysis. The spirocyclic structure developed here is new, and the structural properties of corresponding ligands distinct from the BINOL or SPINOL-based ones, underscoring their great potentials in catalytic applications. The synthetic method is practically useful, as readily available starting materials and an easily accessible catalyst are employed and the reaction is scalable. The work is further strengthened by detailed mechanistic studies and several applications of the developed structure. Overall, this is an impressive work with high novelty as well as great potential utilities, so it is recommended for publication in Nat. Commun.

A minor point: The C₁₀F₇ group in the catalyst is crucial for the enantiocontrol, as the C₁₀F₇-substituted catalyst (D4, E4) significantly boosted the enantioselectivity from 0-7% ee to 80-90% ee when comparing to the analogous catalysts (D1-D3, E1-E3). Comments on these key results might be added in the manuscript.

Response: Thanks for the suggestions, we have updated comments on these key results in the manuscript. More specifically, the following sentences have been added:

*“The C₁₀F₇ group in the catalyst is crucial for the enantiocontrol, as the C₁₀F₇-substituted catalyst (**D4, E4**) significantly boosted the enantioselectivity from 0-7% ee to 80-90% ee when comparing to the analogous catalysts (**D1-D3, E1-E3**).”*

Reviewer #3 (Remarks to the Author):

Chiral scaffolds are indispensable for the progress of asymmetric synthesis, yet the creation of novel, privileged motifs that can more effectively convey asymmetry remains a formidable challenge for chemists. In this manuscript, the authors introduce SPINDOLE, a groundbreaking and versatile chiral framework that ushers in new possibilities for asymmetric synthesis. This innovative approach utilizes readily available and inexpensive indoles and acetone to offer an efficient and cost-effective pathway to enantiopure, axially spirocyclic scaffolds based on indoles. The methodology enables rapid access to SPINDOLEs with good to excellent yields and exceptional enantioselectivities, while demonstrating remarkable compatibility with a wide range of functional groups. Encouraged by these promising results, the authors have embarked on the development of a comprehensive library of catalysts and ligands based

on this unique axially spirocyclic backbone. The SPINDOLE scaffold holds significant potential as a robust platform for designing organocatalysts and ligands, offering a complementary alternative to the widely used BINOL and SPINOL frameworks. Overall, this work presents a powerful strategy for advancing asymmetric catalysis by enabling the creation of versatile chiral frameworks with broad synthetic potential. I recommend the publication of this manuscript in Nature Communications after addressing the following minor issues.

1. As shown in Fig. 3, indole with a -F group at C4 delivers the desired product 4y with high enantioselectivity. In contrast, indole with a -F group at C7 produces the desired product 4z with low enantioselectivity and yield. What could account for this difference?

Response: We hypothesize that the 7-substituent on indole is positioned closer to the catalytic center than other substituents, disrupting key noncovalent interactions with the catalyst and leading to reduced enantioselectivity and yield. This effect may be a hallmark of BINOL-derived Brønsted acid catalysis. While not related to this specific reaction, related studies on TIPSY-based chiral phosphoric acids (CPAs) have shown that bulky silyl groups on the catalyst can completely shut down reactivity (<https://pubs.acs.org/doi/10.1021/jacs.6b02825>). Additionally, modifying substituents near the reaction site on the substrate has been shown to significantly influence reaction outcomes (<https://pubs.acs.org/doi/full/10.1021/jacs.9b11658>). Our observations are consistent with these trends.

2. Please re-draw the structures of products 4u-4x (the Cl, Br, and Me groups are not in the proper positions).

Response: Thanks for the suggestions. We have re-drawn the structures of products 4u-4x. Please see new Fig. 3. Also included here for convenience.

3. The Tan's work related to asymmetric synthesis of spiro-bisindoles should be highlighted in the introduction with a dedicated figure, rather than merely being cited in reference 13.

Response: Thanks for the suggestions. The figure and introduction has been updated. Please new figure 1.

The following sentence has been added to the introduction:

“During the course of our study, the Tan and Sun group reported strategies as applied to SPINOLs to prepare spiro-bisindoles respectively (Fig. 1D), highly supporting our envisioned approach (Fig. 1E).”

4. The footnote of Figure 4 contains numerous formatting errors. Please address and correct these issues.

Response: Thanks for the suggestions. We have carefully checked the formats and corrected the errors.

*This now appears as: “**Fig. 4 Elaboration of SPINDOLE to generate a diverse set of chiral ligand and organocatalyst structures.** Conditions: (a) **4a** (1.0 eq), NaH (3.0 eq), DMF, 0 °C, 1 h; PhNCO (**5a**)/PhNCS (**5b**)/3,5-(CF₃)₂C₆H₃-NCS (**5c**) (3.0 eq), rt, 16 h. (b) **4a** (1.0 eq), *n*BuLi (2.2 eq), THF, 0 °C, 1h; Ph₂PCl or Xyl₂PCl (2.2 eq), rt, 16 h. (c) **4a** (1.0 eq), *n*BuLi (1.1 eq), THF, -78 °C, 1 h; Ph₂PCl (1.1 eq), rt, 2 h. (d) **4a** (1.0 eq), NaH (3.0 eq), DMF, 0 °C, 1 h; 2-(chloromethyl)-4,5-dihydrooxazole (**9a**) or (*R*)-2-(chloromethyl)-4-ethyl-4,5-dihydrooxazole (**9b**) (3.0 eq), rt, 16 h. (e) **4a** or **4d** (77% ee) (1.0eq), NaCNBH₃ (10.0 eq), HOAc (10.0 eq), DCM, 0 °C, 4 h. (f) **4a** (1.0eq), B₂pin₂ (2.0 eq), [Ir(cod)(OMe)]₂ (1.5 mol%), dtbpy (3.0 mol%), THF, reflux, 24 h. (g) **7a** (1.0 eq), Pd(PhCN)₂Cl₂ (1.0 eq), Benzene, 16 h.*

(h) **7a** (1.0 eq), H₂O₂ (10.0 eq), DCM, rt, 1 h. (i) **8** (1.0 eq), NaH (2.0 eq), DMF/THF (v/v=1/4), 0 °C, 0.5 h; 3,5-(CF₃)₂C₆H₃-NCS (**5c**) (2.0 eq), rt, 3 h. (j) **8** (1.0 eq), H₂O₂ (5.0 eq), DCM, rt, 1 h. (k) **16** (1.0 eq), *n*BuLi (1.1 eq), THF, -78 °C, 1h; Ph₂PCl (1.1 eq), rt, 16 h.”

Reviewer #4 (Remarks to the Author):

In the present manuscript, Lai, List, and Reid report an efficient method for the catalytic asymmetric synthesis of spirocyclic bis-indole compounds using indoles and acetone as simple starting substrates. The reaction system, conditions, and chiral Brønsted acid organocatalysts were carefully optimized to afford the target spiro bis-indole products with high enantioselectivities and good yields. The multistep reactions were well-controlled using a BINOL-derived iminoimidodiphosphoric acid catalyst.

Furthermore, the derivatization and application of spiro bis-indole derivatives as chiral ligands were demonstrated. The excellent performance of spiro bis-indole-derived chiral phosphine ligands was elegantly showcased in transition-metal-catalyzed asymmetric transformations. Additionally, a spiro bis-indole-derived chiral amine compound proved to be an effective organocatalyst in asymmetric conjugate addition.

Mechanistic studies were also conducted to elucidate the reaction mechanism of this catalytic asymmetric method for the synthesis of spiro bis-indoles. The manuscript is well-organized, and both the main text and supporting information are meticulously prepared. Key references are appropriately introduced and cited. For these reasons, the reviewer recommends that this manuscript be published in Nature Communications without modifications.

Minor suggestion: A very recent publication on the asymmetric synthesis of related spiro bis-indole compounds should be cited: Angew. Chem. Int. Ed. 2025, e202424773.

Response: Thanks for the suggestion, we have added the reference in the manuscript. Please see citation 18.